# Evolution of multicellular life cycles under costly fragmentation

**Yuriy Pichugin** *, **Arne Traulsen**

Max Planck Institute for Evolutionary Biology, August-Thienemann-Str. 2, 24306 Plön, Germany

* pichugin@evolbio.mpg.de

## Abstract

A fascinating wealth of life cycles is observed in biology, from unicellularity to the concerted fragmentation of multicellular units. However, the understanding of factors driving their evolution is still limited. We show that costs of fragmentation have a major impact on the evolution of life cycles due to their influence on the growth rates of the associated populations. We model a group structured population of undifferentiated cells, where cell clusters reproduce by fragmentation. Fragmentation events are associated with a cost expressed by either a fragmentation delay, an additional risk, or a cell loss. The introduction of such fragmentation costs vastly increases the set of possible life cycles. Based on these findings, we suggest that the evolution of life cycles involving splitting into multiple offspring can be directly associated with the fragmentation cost. Moreover, the impact of this cost alone is strong enough to drive the emergence of multicellular units that eventually split into many single cells, even under scenarios that strongly disfavour collectives compared to solitary individuals.

**Data Availability Statement:** The code and data used in the manuscript are publicly available at https://github.com/yuriypichugin/Costly-fragmentation-life-cycles.

**Funding:** The authors received no specific funding for this work.

## Author summary

Even among the simplest bacteria, there is an impressive diversity in reproduction modes: Some organisms split their bodies into multicellular pieces—others produce unicellular propagules. Some organisms give rise to multiple offspring at once, while others fragment into only two parts. What drives the evolution of such reproduction modes? Here, we theoretically investigate a previously overlooked factor: the costs caused by the fragmentation event itself. We show that reproduction costs can be responsible for the evolution of fragmentation into multiple parts. However, not any fragmentation mode is possible—many modes cannot evolve under any cost. Based on mathematical reasoning alone, we can thus make general statements about the evolution of life cycles. Since our results demonstrate that the evolution of reproduction modes is heavily influenced by the costs of the reproduction act, they call for a more thorough experimental consideration of these costs.

**Competing interests:** The authors have declared that no competing interests exist.

## Introduction

All living and evolving organisms are born, grow and reproduce, giving birth to new organisms [1–10]. This cycle is central to the existence of life on Earth, as natural selection promotes species which perform this cycle in a more efficient way than others. Surprisingly, even presumably simple organisms demonstrate a great variety of life cycles: *Staphylococcus aureus* produces unicellular propagules [11], cyanobacteria filaments fragment into multicellular threads [12], bacterial biofilms perform seeding dispersal, in which a biofilm composed of sessile cells develops cavities filled with motile cells later released into the environment [13, 14]. Which factors drive the evolution of life cycles and fragmentation modes remains largely unknown, even for the simplest multicellular species. Nevertheless, the examples above illustrate that there is no universally optimal fragmentation mode. Instead, the fragmentation mode is an adaptation to the environmental conditions limited by the biological constraints of the organism [9, 15–18].

One such constraint, which can have an impact on the evolution of life cycles, is the fragmentation cost. There is substantial evidence that fragmentation is costly in natural populations. For example, during the fragmentation of simple multicellular organisms, the release of cells requires breaking the cellular matrix, which takes time and resources [19, 20]. Also, not every cell may pass to the next generation. For instance, in slime molds some cells form a stalk while others serve as spores. The stalk cells die shortly after the spore cells are released [21]. In the case of *Volvox carteri* colonies, cells constituting the outer layer of the colony die upon release of the offspring [22]. In multicellular colonies of *Saccharomyces cerevisiae* evolved in settling experiments [23, 24], fragmentation events are facilitated by the apoptosis of inner cells, which weakens the links stabilizing a tree-like structure of a colony. Combined, there is evidence that fragmentation can be associated with a conspicuous cost.

Whether the fragmentation cost is an actual driving force of evolution in natural populations is an open experimental question. Nevertheless, our previous model predicts that costless fragmentation promotes the evolution of only binary fragmentation modes [25]. This theoretical finding thus excludes the evolution of life cycles involving fragmentation into multiple parts—but this is a common reproduction strategy among simple multicellular organisms found in nature [26–30], see Fig 1.

There are only a few theoretical studies of the evolution of reproductive modes which explicitly take into account the fragmentation cost. Libby et al. modelled the evolution of life cycles of colonial forms of *S. cerevisiae* [31]. In their model, the fragmentation of tree-structured cell clusters was attributed to the death of cells. These cells become weak links and loose connections with neighbouring cells, causing the fragmentation of the cluster. However, while Libby et al. considered a detailed model of a binary fragmentation of cell clusters, they did not investigate the whole range of fragmentation outcomes. In our own previous work, we have exhaustively analysed all possible ways of fragmentation and found evolutionarily optimal life cycles in various environments [25]. For costless fragmentation, only binary fragmentation, where a cell cluster splits into two parts, can be evolutionarily optimal in terms of maximising population growth. The same holds for the case of proportional cost, where upon division into $s$ parts, $s - 1$ cells die. However, for fragmentation with a fixed cost in the form of a single cell loss, fragmentation modes with multiple offspring can become evolutionarily optimal [25].

In this study, we investigate the influence of fragmentation cost on the evolution of "staying together" life cycles [32]. Building upon the framework of [25], we explicitly incorporate fragmentation costs arising from three scenarios: fragmentation delay, fragmentation risk, and cell loss. We identify evolutionarily optimal life cycles maximizing the population growth rate. We show that some life cycles are never optimal and show mathematically why there are such

**A**

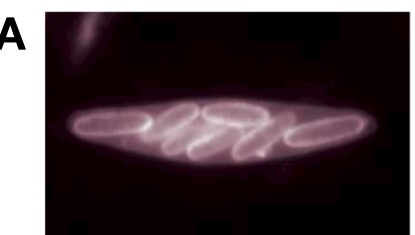

*Metabacterium polyspora*

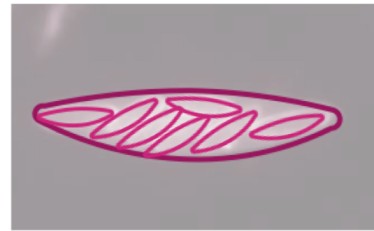

Several cells grow inside a maternal cell

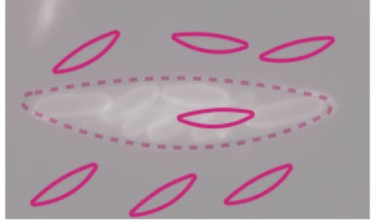

Several cells are released by lysis of the maternal cell

**B**

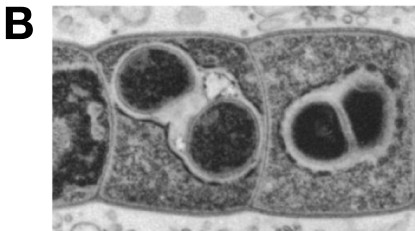

Segmented filamentous bacteria

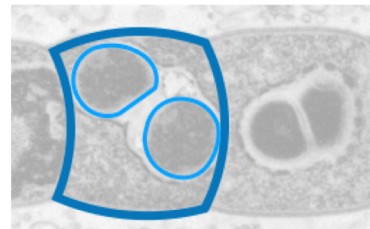

Two holdfast-bearing cells grow inside a maternal cell

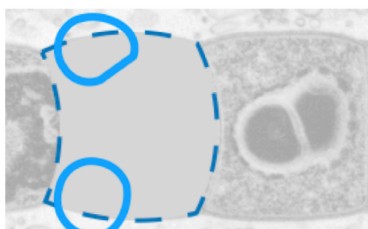

Two cells are released by lysis of the maternal cell

**C**

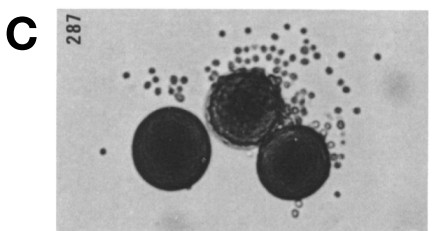

cyanobacteria *Stanieria*

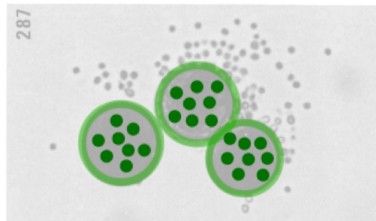

Many cells grow inside an extracellular matrix

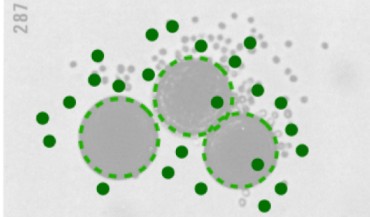

Extracellular matrix is discarded and cells are released

**Fig 1. Examples of multiple fragmentation in nature and their interpretation in terms of our model.** (**A**) *M. polyspora* grows multiple endopsora, released after the maternal cell lysis (picture adopted from [27], Copyright (1998) National Academy of Sciences). (**B**) segmented filamentous bacteria grows two holdfast-bearing cells inside a maternal cell. These cells are released in the result of the maternal cell lysis (picture adopted from [29]). The death of maternal cells is the fragmentation cost in these two examples. (**C**) genus *Stanieria* grows multiple cells within a single extracellular matrix. These cells are released simultaneously upon the break of the matrix (Picture adopted from [30]). The loss of the extracellular matrix corresponds to a fragmentation cost here.

"forbidden" life cycles. Then, we investigate which life cycles are more likely to evolve in the context of costly fragmentation and show that binary and (nearly) equal split life cycles are by far the most abundant after population growth rate optimization. Finally, we consider in detail environments in which an increase in cluster size always reduces the performance of the cluster, i.e. the fastest growth and the best protection is achieved by independent cells. We show that even in these environments that strongly disfavour collective living, fragmentation costs can promote the evolution of life cycles involving the emergence of multicellular units that later split into individual cells. Therefore, fragmentation costs, previously overlooked, are likely to be a notable factor of life cycle evolution.

## Model

Here, we first discuss the development of multicellular units by dividing cells staying together. Then, we present how fragmentation of units can occur. and how the fragmentation costs are taken into account. After that, we introduce resource limitation that restricts the population

size. Finally, we discuss the selection occurring in the population defined by the processes above.

## Development and death of units

We consider a growing population composed of unstructured units of identical cells, which emerge, increase in size, and fragment into offspring units, thus completing a life cycle. A unit's size increases due to dividing cells staying together after a cell division [32]. Due to the absence of any internal structure, the properties of a unit are determined by its size $i$ alone. The minimal size of a unit is $i = 1$ (a solitary cell) and the maximal size is $i = n$. We denote the abundance of units of $i$ cells in the population as $x_i$. Units of size $i$ have death rate $d_i$, and cells in such a unit have division rate $b_i$. Thus the size of a unit increases with the rate $ib_i$, see Fig 2A. In line with recent studies on the experimental evolution of multicellularity [7, 23, 33, 34], we assume that the vectors of division rates $\mathbf{b} = (b_1, \ldots, b_n)$ and death rates $\mathbf{d} = (d_1, \ldots, b_n)$ mainly come from the interaction of units with external factors, such as the availability of nutrients, the presence of hazards, etc. Hence, we refer to the pair $(\mathbf{b}, \mathbf{d})$ as *environment*.

## Fragmentation

In each investigated life cycle, units increase in size by means of cell division up to size $m$ called *maturity size* (in our model, we have $m \le n$). Maturity sizes for different life cycles may be different. If a unit of size $m$ survives until the next cell division, it will immediately fragment into smaller units according to the specific pattern, unique for each life cycle. As any unit can be characterized by the number of cells comprising it, any fragmentation event can be characterized by a partition of this integer number, see Fig 2B. A partition is a way of decomposing an integer $m$ into a sum of integers without regard to order, and summands are called parts [35].

As an example, consider life cycles with maturity size $m = 3$. There, the maximal size of a unit is three cells and the unit fragments when a fourth cell emerges. There are four different ways how four cells can be distributed between two or more units. First, the parental unit can fragment into two unequal pieces of three cells and one cell (partition 3+1 of 4). Second, the parental unit can fragment into two equal pieces of two cells each (partition 2+2 of 4). Third, the parental unit can fragment into three pieces, one of which has two cells and two others are unicellular (partition 2+1+1 of 4). Finally, the parental unit can split into four solitary cells (partition 1+1+1+1 of four), see Fig 2B for graphical illustration of these fragmentation modes. For simplicity of notation, we refer to life cycles by their partitions. For example, the unicellular life cycle, where cells immediately fragment after cell division, is called "1+1 life cycle".

The number of partitions of $m$ grows quickly with $m$. In the numerical part of the current study, we use a maximal size of $n = 19$ and thus size before fragmentation does not exceed 20, which leads to 2693 different life cycles [25]. Our analytical results, however, are valid for any choice of $n$.

In the course of this manuscript we only consider deterministic life cycles, where all groups fragment according to the same fragmentation mode. It can be shown that stochastic life cycles, representing bet-hedging reproductive strategies, are always evolutionarily suboptimal in the scope of our model, see S6 Text. Stochastic life cycles can evolve, however, when the environment $(\mathbf{b}, \mathbf{d})$ is variable in time [36].

## Three ways of implementing fragmentation costs

Next, we assume that the fragmentation of a unit comes at a cost for this unit. By cost, we mean that comparing two identical units, one of which fragments at some moment, while another continues growing, the expected size of the fragmenting unit (or the sum of the

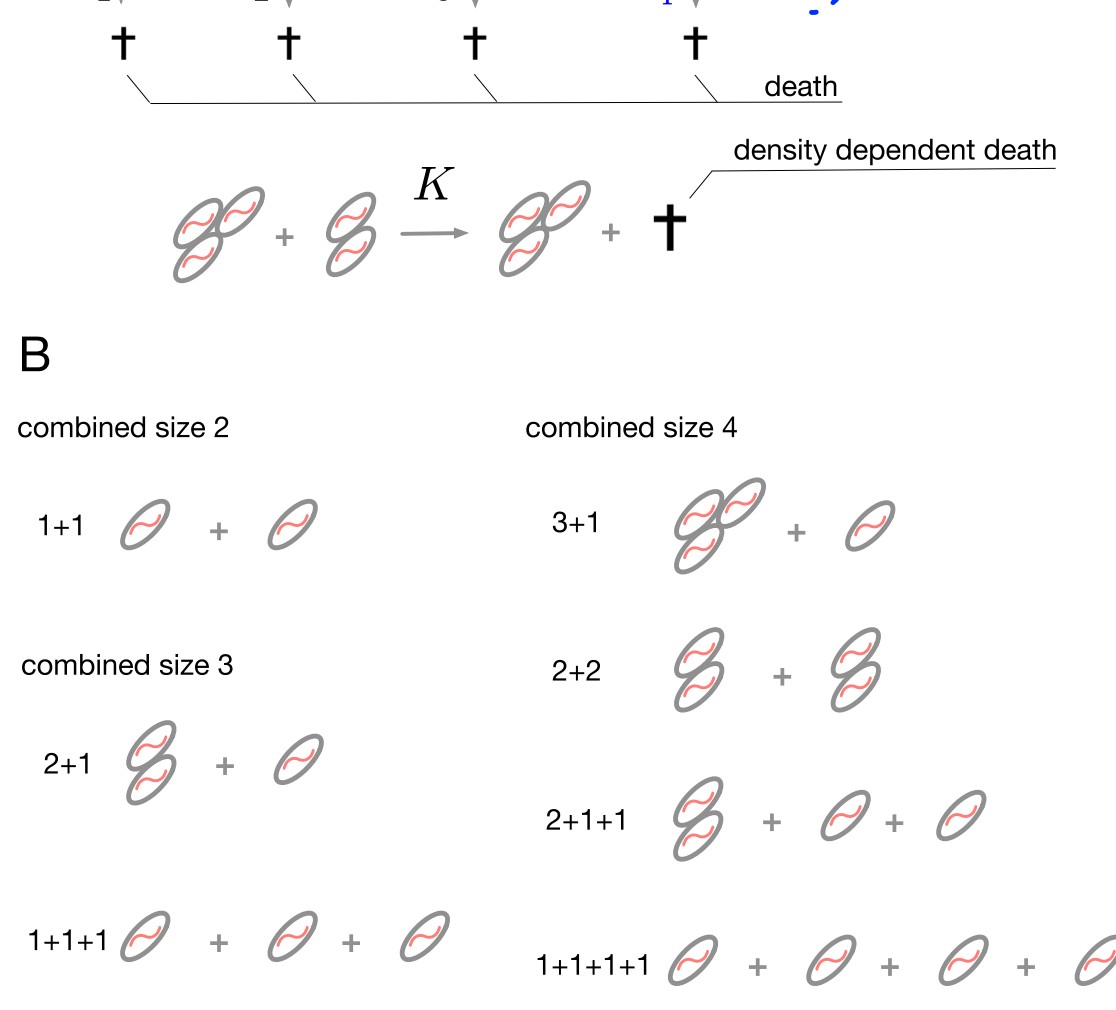

**Fig 2. Model of life cycles.** (**A**) Multicellular units increase in size and fragment to produce new units. Cell division ($b_i$) and unit death ($d_i$) rates depend on the size of the unit and are determined by the environment. If the fragmentation process is costly, the division rate at the maturity size may be smaller than prescribed by the environment alone $b'_m \leq b_m$ (see Eq (1)), the death rate at the maturity size may be larger than prescribed by the environment alone $d'_m \geq d_m$ (see Eq (2)), and some cells may be lost upon the fragmentation (highlighted in blue). In this example, $m = 4$, so the unit fragments when it reaches five cells, one cell is lost in fragmentation and the remaining four are split by the fragmentation mode 2+2. (**B**) The fragmentation mode of cell clusters can be described by a sum of integers. All possible fragmentations of units of size 2, 3, and 4 are presented here. Different life cycles have different growth rates and we are looking for the fastest growth in this context.

expected sizes of its offspring) will be smaller for at least a brief period of time. We consider three qualitatively different scenarios that describe a fragmentation cost: fragmentation delay, fragmentation risk, and fragmentation loss. Fragmentation costs represents various physiological restrictions of the units life history and assumed to be independent on the life cycle. Thus, growth competition between different life cycles maximizes the population growth rate, given the fixed environment and the fixed fragmentation cost.

**Fragmentation delay.** In the case of fragmentation delays, the process of fragmentation is not immediate and takes time $T$. This scenario covers situations where the fragmentation of a unit requires the investment of resources, which would otherwise be spent on cell propagation. The transition time is inverse to the transition rate, thus we define the rate of fragmentation $b'_m$ of a unit of size $m$ by

$$\frac{1}{mb'_m} = \frac{1}{mb_m} + T, \quad \text{such that} \quad b'_m = \frac{b_m}{1 + mb_m T} \leq b_m, \tag{1}$$

where $T$ is the fragmentation delay.

**Fragmentation with risk of death.** In the case of fragmentation with risk, a unit has an increased death risk prior to fragmentation. For example, a unit could leave the shelter or break its shell in order to reproduce. Under this scenario, the death rate at the maturity size $d'_m$ is increased

$$d'_m = d_m + R, \tag{2}$$

where $R$ is the increase in the death rate triggered by the additional risk.

**Fragmentation with loss.** For fragmentation with loss, $L$ cells die just before a unit fragments. Thus, the combined size of offspring units is $L$ cells smaller than the size of the parent unit. In this scenario, the fragmentation following the increment of size from $m$ cells to $m + 1$ cells is characterized by a partition $\kappa$ of $m + 1 - L$. We assume that $L$ is independent of the partition chosen. Thus, the minimal maturity size possible is $m = 1 + L$ and the maximal possible combined offspring size is $n + 1 - L$.

Note, that the three considered scenarios are not mutually exclusive, all three types of cost may be present simultaneously. However, for simplicity we do not consider their interplay in the current work.

## Resource limitation

With constant growth and death rates, the population either goes extinct or exponentially increases in size. This does not occur in natural population due to the limitation of resources such as nutrients, physical space, or sunlight. To represent this constraint, we now add a density dependent death process to our model. The rate of unit death is directly proportional to the total number of units in a population with the reaction constant $1/K \ll 1$, see Fig 2A. In the long run, a population subjected to such process reaches the carrying capacity, with a magnitude of $K$ units.

## Life cycle competition

The dynamics of population is governed by the growth of units, costly fragmentation into smaller parts, and resource competition. For a given life cycle, the state of a population can be described by the abundances of units $x_i$ of each possible size $i$ from one cell to $m$ cells, given by the vector $(x_1, x_2, \cdots, x_m)$. There are no units of size $m + 1$ or larger, because any unit fragments immediately after the next cell arises in a unit of the maturity size $m$.

The dynamics of the population state can be expressed as a system of $m$ differential equations—one equation for each unit size. The set of equations for a life cycle with fragmentation pattern $\kappa$ is given by

$$\frac{d}{dt}x_1 = -b_1 x_1 - d_1 x_1 + \pi_1(\kappa) m b'_m x_m - \frac{1}{K} X x_1 \tag{3a}$$

$$\frac{d}{dt}x_i = \underbrace{-i b_i x_i + (i-1) b_{i-1} x_{i-1}}_{\text{Cell division}} \underbrace{-d_i x_i}_{\text{Death}} + \underbrace{\pi_i(\kappa) m b'_m x_m}_{\text{Fragmentation}} \underbrace{-\frac{1}{K} X x_i}_{\text{Competition}} \quad \text{for} \quad 1 < i < m \tag{3b}$$

$$\frac{d}{dt}x_m = -m b'_m x_m + (m-1) b_{m-1} x_{m-1} - d'_m x_m + \pi_m(\kappa) m b'_m x_m - \frac{1}{K} X x_m, \tag{3c}$$

where $X = \sum_{j=1}^{m} x_j$ is the total number of groups in the whole population, including lineages with other life cycles.

Here, Eqs (3a) and (3b) describe the dynamics of the abundances of units $x_i$ that increase in size without fragmentation, because they do not reach the maturity size $m$. The first two terms in Eq (3b) $-i b_i x_i + (i-1) b_{i-1} x_{i-1}$ describe the change in $x_i$ due to cell division. The third term $-d_i x_i$ describes the death of units. The next term $\pi_i(\kappa) m b'_m x_m$ describes the emergence of new units of size $i$ resulting from the fragmentation of mature units. The integer $\pi_i(\kappa)$ is the number of units of size $i$ that emerge in a single act of fragmentation according to the partition $\kappa$ (for example for $\kappa = 1 + 1 + 1$, we have $\pi_1(\kappa) = 3$ and $\pi_i(\kappa) = 0$ for $i > 1$), and $m b'_m$ is the growth rate prior to fragmentation (see Eq (1)). Finally, the last term $-\frac{1}{K} X x_i$ describes competition, leading to density dependent death.

Eq (3c) describes the dynamics of units of maturity size $m$, which will inevitably fragment according to the partition $\kappa$ upon the next cell division. For fragmentation with delay, the rate of transition to the next state (fragmentation) is smaller than the division rate determined by the environment ($b'_m < b_m$, see Eq (1)). For fragmentation with risk, the death rate is larger ($d'_m > d_m$) than induced by the environment death rate vector $\mathbf{d}$ (see Eq (2)).

Thus, Eq (3) describe a single life cycle. It can be characterized by the growth rate of the life cycle, which emerges as the exponential growth rate of the lineage performing this life cycle once the relative sizes of each unit does not change anymore. This exponential growth rate is the key parameter for evolutionary competition in our model: We can imagine a population, in which different lineages perform different life cycles. There, some lineages will become more abundant, while others will go extinct. We investigate which factors provide an evolutionarily advantage to a life cycle to win the competition by natural selection.

## Results

### In any environment there is a single evolutionarily optimal life cycle

Our first finding is that the competition of life cycles has the same outcome as in the simplified model without resource limitation ($K = 0$), see S1 Text for details. Without resource limitation, the system is described by a linear matrix population model characterized by the matrix $A_{ij}$,

$$\frac{d}{dt}x_i = \sum_j A_{ij} x_j, \tag{4}$$

see S2 Text for details of the matrix construction. There, the competition of life cycles is determined by their intrinsic growth rate, equal to the leading eigenvalue of the matrix $A_{ij}$.

Consequently, in any environment, the life cycle having the maximal growth rate will eventually dominate the system, independently of the initial conditions.

Below, to find the evolutionarily optimal life cycle, we fix the division rates (**b**), the death rates (**d**), and the fragmentation costs ($T$, $R$, $L$). Then, we compute the population growth rates for each individual life cycle and find one with the maximal value. Our analytical results on the forbidden life cycles are valid for any maximal size $n$ and any combination of environment and fragmentation costs. For simplicity, our numerical results consider each of the three fragmentation cost scenarios separately.

## Some life cycles cannot be evolutionarily optimal under any environment

Here, we show that many life cycles cannot be evolutionary optimal in any environment. We label these life cycles "forbidden life cycles". Consequently, we call a life cycle that can be evolutionarily optimal under some environment an "allowed life cycle".

All three scenarios of the fragmentation cost (delay, risk and loss) lead to the same condition for a life cycle to be allowed: fragmentation by the partition $\kappa$ leads to an allowed life cycle only if any two subsets of its parts with the same sum have the same form. Consequently, $\kappa$ leads to a forbidden life cycle if $\kappa$ contains two subsets of parts with the same sum, but with a different form. In other words, within the partition $\kappa$, we can find two different partitions $\tau_1$ and $\tau_2$ of the same integer $j$. For any environment and any fragmentation cost scenario, every forbidden life cycle has smaller population growth rate $\lambda$ than at least one allowed life cycle, see S3 Text for a proof.

The simplest example of a forbidden life cycle is the partition 2+1+1, which has two different offspring subsets: 2 and 1+1, both having the same combined size 2. The fate of the two subpopulations founded by the offspring, 2 and 1+1, is entirely independent of each other. Generically, either the subpopulation starting with a unit of two cells or the subpopulation starting with two unicells will have the larger growth rate. In the first case, the life cycle 2+2 would ultimately lead to a larger growth than 2+1+1, whereas in the second case the life cycle 1+1+1+1 would lead to a larger growth. More examples of forbidden life cycles are presented on Fig 3A. The proportion of forbidden life cycles increases rapidly with the partition sum (see black bars on Fig 3B). Computationally assessing each of our 2693 partitions, we found 2006 partitions corresponding to forbidden life cycles, roughly 75%.

## Classification of life cycles

The total amount of allowed life cycles is still too large to track each of them individually. Therefore, we distinguish three classes: binary fragmentations, (nearly) equal splits, and seeding life cycles, see also Fig 3A.

Binary fragmentations have only two parts and are of the form $\kappa = a + b$. Examples of binary partitions are 2+2 and 7+1. Among the non-binary fragmentation modes (where we have at least three offspring units), we distinguish (nearly) equal splits and seeding partitions. Note that we call the split into two identical units binary (but at the same time, it would be an equal split). In (nearly) equal split partitions, the sizes of the largest and the smallest offspring differ by no more than one cell. In other words, (nearly) equal splits have the form $\kappa = a + \ldots + a + b + \ldots + b$ such that $b = a$ (true equal split) or $b = a - 1$ (nearly equal split). For (nearly) equal split partition to be allowed, there must be either less than $b$ parts of size $a$, or less than $a$ parts of size $b$. Examples of (nearly) equal splits are 1+1+1 and 3+3+2+2. (Nearly) equal splits represent scenarios where cells are distributed among multiple offspring units as evenly as possible.

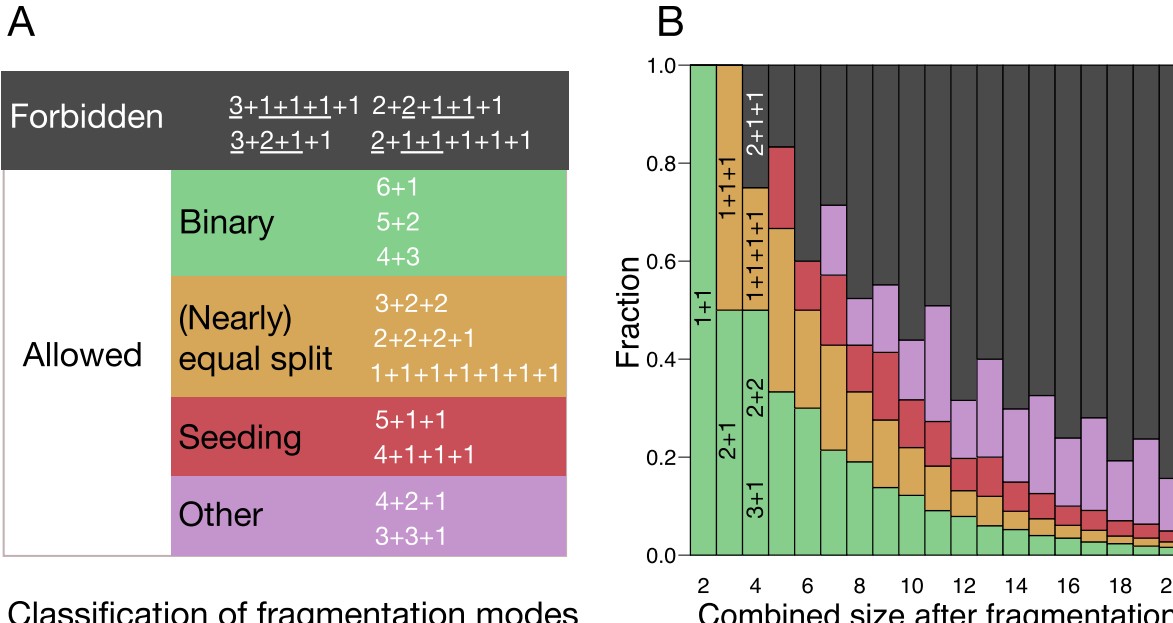

**Fig 3. The majority of life cycles cannot win growth competition and are thus called forbidden.** (A) All possible fragmentation modes with combined fragments size equal to 7. Allowed modes are further broken into binary, (nearly) equal split, seeding, and other classes, according to the definitions in the main text. For each of the forbidden modes, a couple of different subsets of parts with the same sum are underlined. (B) Proportions of different classes of fragmentation modes for different combined size at fragmentation. For sizes 2 and 3, all partitions are allowed. Starting from size 4, some are forbidden (for 4, it is 2+1+1). The proportion of forbidden modes grows rapidly with the size. For even sizes, more forbidden modes exist. Among the allowed modes, the proportions of binary, (nearly) equal split and seeding classes rapidly declines.

Finally, seeding partitions have the form $\kappa = a + b + \ldots + b$ with the restriction $a > b + \ldots + b$ to make sure that the partition is allowed. Examples of seeding are 3+1+1 and 7+2+2+2. Distinguishing the seeding fragmentation modes is inspired by seeding dispersal exhibited by biofilms, where a small portion of cells leaves the parent cell cluster in an act of fragmentation.

Some allowed partitions do not belong to any of these three classes, for instance, 4+2+1 and 3+3+1. We lump them together in the "other" partitions class. The proportion of binary, (nearly) equal, and seeding partitions among all allowed partitions decreases with the partition sum, see Fig 3B). For $n = 19$, we have 2693 partitions in total. Among the 687 allowed partitions, there are 100 binary partitions, 80 (nearly) equal split partitions, 110 seeding partitions and 397 other allowed partitions, which do not belong to either of these three classes.

## (Nearly) equal split and binary fragmentation life cycles are overrepresented for random environments

Our previous findings introduced the range of potentially optimal life cycles, but they did not give any insight into which life cycles are more likely to be optimal. To address this, it would be necessary to know the environment (profiles **b** and **d**) and the fragmentation costs ($T$, $R$, $L$). Unfortunately, such empirical data is not available, neither for natural nor for experimental populations.

Thus we generated a set of 10000 random environments, see S4 Text. We independently assessed each of the three fragmentation cost scenarios (delay, risk, or loss). For delay and risk, we used 200 different values of $T$ and $R$, respectively, in the interval [0, 20]. For the scenario of the cell loss, we used each possible value of $L$ from 0 to $n - 2 = 18$. For each combination of the

environment from the random set and the fragmentation cost, we numerically computed the evolutionarily optimal life cycle—in total more than $4 \cdot 10^6$ data points. This allows to screen through an extremely diverse set of selective conditions. Using this approach, we can distinguish life cycles remaining evolutionarily optimal under a broad range of conditions from life cycles which may only evolve in special circumstances.

When the fragmentation is costless ($T = R = L = 0$) only binary fragmentations can be evolutionarily optimal [25], while seeding, equal split, and the other life cycles cannot evolve. This leaves only 14% of the total 687 allowed life cycles. It is remarkable that under a costly fragmentation, more than 96% (662) of life cycles occur as an evolutionary optimal in some combination of the environment and the fragmentation cost for our parameter set. All allowed life cycles with maturity size up to $m = 10$ can be evolutionarily optimal in each of the three scenarios for the fragmentation costs. Additionally, all allowed life cycles with maturity size of up to $m = 14$ can be evolutionarily optimal in at least one of the three scenarios. Allowed life cycles, which were not evolutionarily optimal have large maturity size and likely are optimal under an extremely narrow range of conditions, which was not captured by our screening due to the limited sample size. Our findings suggest that any allowed life cycle can become evolutionarily optimal given the right combination of environment and fragmentation cost.

In the delay scenario, the most frequently observed fragmentation mode is unicellularity, 1 +1, see Fig 4A. Among the ten most frequent fragmentation modes, we observe in particular binary and equal split life cycles at large and small sizes. Thus, delay fragmentation costs provide an advantage to equal split life cycles relative to the costless scenario.

In the death risk scenario, the most frequently observed fragmentation mode is the equal split into $1 + \ldots + 1$ at the maximal possible size $m = 20$, see Fig 4B. Other frequent life cycles are equal splits at a large size (20, 19, 18, etc). Thus, delay fragmentation costs provide a large advantage to equal split life cycles with large maturity sizes relative to the costless scenario.

In the cell loss scenario, the most frequently observed fragmentation mode is unicellularity, 1+1, see Fig 4C. Other frequent life cycles are mostly equal splits at small sizes (3, 4, 5, etc). Altogether, in all scenarios, the ten most frequent fragmentation modes are either binary or equal split. Thus, loss fragmentation costs provide an advantage again to equal split life cycles relative to the costless scenario.

If there are no fragmentation costs ($T = R = L = 0$), only binary partitions are evolutionarily optimal, as shown in [25]. However, even minimal fragmentation costs allow life cycles from all four classes to evolve, see Fig 4D–4F.

Some life cycles are vastly overrepresented: The binary and (nearly) equal split classes of life cycles represent only 26% of all allowed life cycles. Nevertheless, these classes constitute more than 80% of the observed evolutionarily optimal life cycles for any fragmentation cost in any scenario. The fragmentation delay scenario promotes mostly binary fragmentations, while fragmentation risk and loss promote (nearly) equal splits. Qualitatively the same pattern is observed in other environments with correlated birth and death rates profiles: beneficial (birth rate increase with size, while death rate decrease) and unimodal (at the optimal size of 10 cells, birth rate has a maximum, while death rate has a minimum), see S5 Text.

The distribution of classes for fragmentation with losses looks irregular, see Fig 4F. This results from the decrease of the available partition set with increasing $L$. At $L = 0$, all 687 allowed fragmentation modes are available. With increasing $L$, this number drops as the maximal group size is limited to 19. For $L = 1$, only 589 life cycles can be executed. Finally, for $L = 18$, the only possible life cycle is to fragment upon the birth of the 20-th cell into two solitary cells ($\kappa = 1 + 1$), losing the other 18 as a fragmentation cost. Hence, the set of life cycles to choose from is different for each $L$, which explains the seemingly irregular distribution of evolutionarily optimal life cycles.

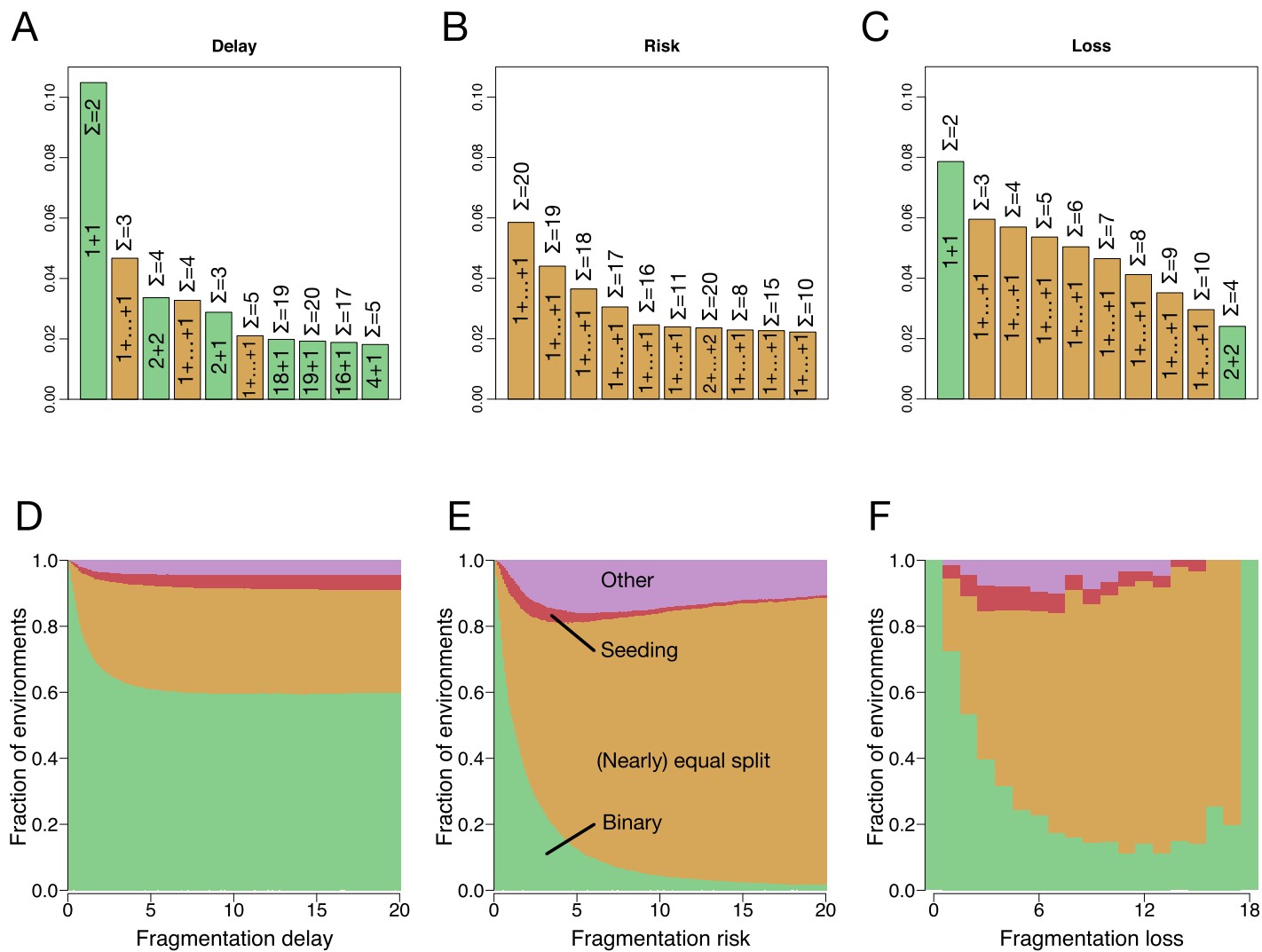

**Fig 4. Fragmentations by binary and (nearly) equal split partitions are likely to evolve in random environments.** The top panels present the ten most frequent fragmentation modes for (**A**) fragmentation with delay, (**B**) fragmentation with increased death risk and (**C**) fragmentation with cell loss, respectively. Each bar shows the frequency of the corresponding life cycle to be evolutionarily optimal. The colour of the bar represents the class of the life cycle, see Fig 3. (Nearly) equal split life cycles are represented in the form of 1+ . . . + 1 and the total number of cells Σ. The bottom panels (**D—F**) present the fractions of each of binary fragmentation, (nearly) equal split, seeding, and other allowed fragmentation modes as functions of fragmentation cost for the same scenarios. The majority of random environments promote the evolution of (nearly) equal split and binary fragmentation modes.

## Fragmentation costs can drive the formation of multicellular units

Multicellular units evolve when the existence of cells in a collective provides some benefit, expressed for example in a form of better resource acquisition or protection from external threats. However, for costly fragmentation, even when the existence in groups is detrimental to cells comprising them, the formation of multicellular units may become evolutionarily beneficial. We have constructed a set of 10000 random detrimental environments in which the death rate increases monotonically with the size of the unit, while the division rate monotonically decreases with the size of the unit (see S4 Text). Consequently, for costless fragmentation, the optimal life cycle for all detrimental environments is unicellular, i.e. uses the partition 1 + 1. However, under costly fragmentation, other—multicellular—life cycles become optimal, see Fig 5.

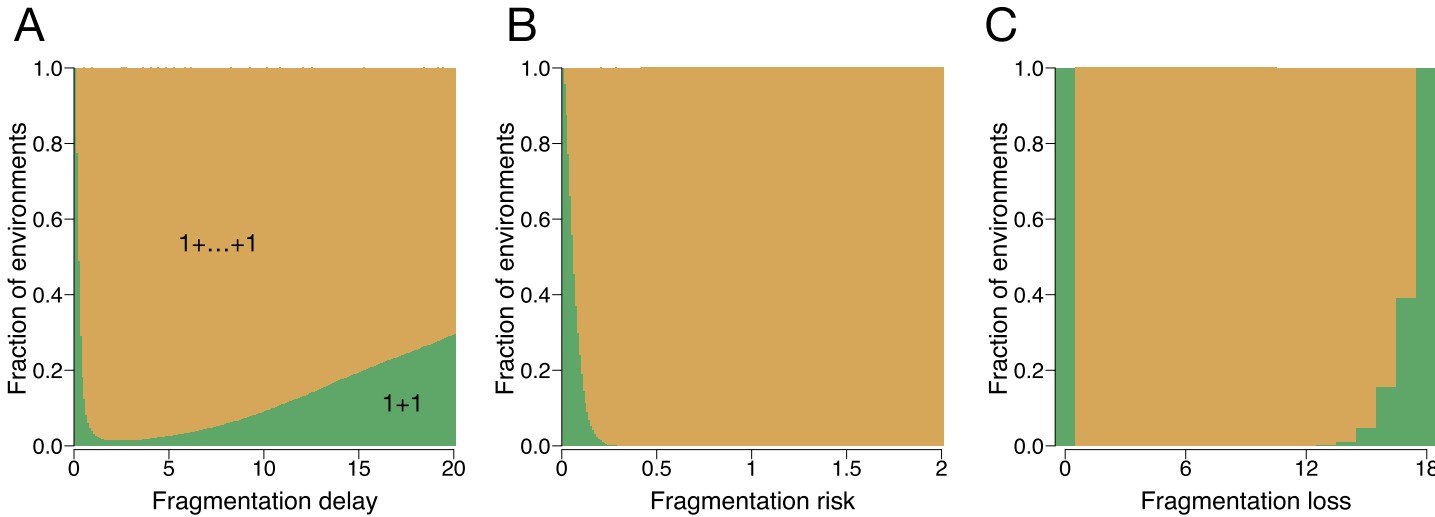

**Fig 5. Multicellular units can evolve in detrimental environments.** Each panel presents the fraction of the minimal, binary life cycle 1+1 (green) and equal split life cycles with a larger maturity size (orange) for random detrimental environments, where division rates decrease and death rates increase with cluster size. Panel (**A**) shows fragmentation with delay, (**B**)—fragmentation with risk, and (**C**)—fragmentation with cell loss. The increase in fragmentation costs, drives the evolution of life cycles involving formation of multicellular units. All evolutionarily optimal life cycles found have the form 1+ . . . + 1.

Two factors contribute to this effect: First, it takes more time to reach a larger size and this makes fragmentation less frequent, so the cost paid per time unit is smaller. Second, larger unit size at fragmentation makes it possible to share the cost of fragmentation among more units. Therefore, fragmentation costs can establish selection pressures promoting multicellular life cycles strong enough to overcome the impact of the environment.

Note, that in the scenario of fragmentation with loss ($L > 0$), all fragmentation modes, including 1+1, require multicellular units to be produced at an intermediary stage of the life cycle. For this case, the conclusion is slightly different—cell loss promotes life cycles with maturity sizes larger than the minimally possible $m = 1 + L$, see Fig 5C.

We also observed that for all detrimental environments and all scenarios of the fragmentation cost, all observed multicellular life cycles are equal splits in the form $1 + 1 + \cdots + 1$. In these life cycles, initially solitary cells develop into multicellular units, which then turn into solitary cells again. The intuition behind this result is that a solitary cell is the most effective state available to the population in detrimental environments in terms of division rate and death risk. Therefore, multicellular life cycles passing through the unicellular state have an advantage over life cycles, where groups fragment into larger pieces.

## Discussion

In order to reproduce, an organism undergoing a clonal development has to partition its body into several pieces. This is true across all levels of biological organization, from the fragmentation of cyanobacteria filaments to the birth act in viviparous animals. However, the particular implementation of this process drastically varies between species. Even in bacteria, we observe a great variety of reproduction patterns among simple multicellular species, where reproduction occurs by means of simple fragmentation [11–13, 23, 26]. What drives the evolution of these fragmentation modes is an open question. In this study, we constructed a matrix population model of multicellular units that increase in size by staying together. In our model, fragmentation costs result in less immediate expected biomass than an increase in the size of a unit without fragmentation. We considered three scenarios of fragmentation costs: delay (reduced

rate of the cell division), death risk (increased chance of the unit death at fragmentation), and cell loss (guaranteed loss of some biomass), see Fig 2. We numerically assessed all possible fragmentation modes available to units up to 20 cells and identified those modes leading to a maximum population growth rate.

The roots of our approach come from the field of demography, which studies age- or stage-structured populations [37]. Our model utilizes parameters typical for demographic models: rates of transitions between states and mortalities at each stage. As of today, the overlap between demography (our method) and microbiology (our subject of study) is almost non-existent and empirical data are just not available yet. Nevertheless, the methodology to gather all data needed for models of our type exists and is already developed. Once a sub-field of the demography of simple multicellular species will emerge, our model can naturally be adapted to the new experimental paradigm. Predictions from our model are thus ready to be tested once experimental data are available. However, we expect that such experiments will also naturally trigger the further development of theoretical models.

Many of our results are valid for an arbitrary choice of parameters' values and do thus not rely on the availability of experimental data, see Fig 3. For other results, we used a large set of random parameters. This set established an extremely diverse manifold of parameter values—which can be expected to be comparable or even exceed the diversity expected in natural populations. This makes it possible to find common patterns among evolutionarily optimal life cycles typical for majority of parameters combinations, including the ones present in nature. For example, the evolution of multicellular life cycles driven by fragmentation costs was observed among more than 99% of random detrimental environments, see Fig 5. Thus, our study suggest that the costs of fragmentation could be a important factor in the evolution of multicellular life cycles.

Our previous findings [25] indicate that under costless fragmentation only binary fragmentation modes can become evolutionary optimal, while a small cell loss $L = 1$ may promote fragmentation into multiple parts with similar sizes. In the present work, we show that the influence of the fragmentation costs is much more substantial. First of all, costly fragmentation allows the evolution of a diverse set of fragmentation modes. These are not limited to the (nearly) equal splits we observed before: For instance, this includes seeding partitions, where a single large offspring unit is accompanied by a number of smaller propagules, conceptually similar to the seeding dispersal in biofilms, see Fig 3A.

At the same time, costly fragmentation still imposes restrictions on which fragmentation modes can evolve. The majority of them cannot evolve under costly fragmentation, see Fig 3B. We found that all three considered scenarios of the fragmentation cost share the same set of forbidden life cycles—these contain two different offspring subsets with the same combined size. Given the difference between the considered cost scenarios, this result is striking.

Among all allowed life cycles, the evolution of life cycles with costly fragmentation generally gives rise to binary or (nearly) equal split fragmentation modes. These two classes constitute only a small fraction of all allowed life cycles, but appeared to be evolutionarily optimal under the vast majority of random environments for all three scenarios of the fragmentation cost, see Fig 4. Looking at natural populations, binary fission seems to be the dominant mode of fragmentation among bacteria and simple eukaryotes [38]. The majority of species, which utilize fragmentation into more than two parts, do so by fission in multiple unicellular propagules [38]. A notable exception is the fragmentation mode of segmented filamentous bacteria [28]. Thus, binary fragmentation and (nearly) equal split are not only promoted by our model, but are also relatively widespread in nature.

Life cycles with single cell bottlenecks have a central role in the life histories of complex animal multicellularity. The fragmentation costs considered in our work promote the evolution of

such reproduction modes: many of multiple fragmentation modes (only able to evolve under costly fragmentation) feature unicellular propagules. Specifically, reproduction via fission into solitary cells in a form $1 + \ldots + 1$ appear among the most frequently optimal life cycles in every scenario of fragmentation costs, see Fig 4A–4C. This demonstrates that reproduction costs open additional opportunities for evolution of life cycles with single cell bottlenecks.

The evolution of cell collectives from unicellular ancestors is often considered to be driven by some ongoing benefits provided by the group membership such as better protection [39], access to novel resources [40] and the opportunity to cooperate (reviewed in [41] and in [42]). In our work, we have shown that such ongoing benefits of being in a group are not a necessary condition for the evolution of collectives. The impact of the fragmentation cost can be strong enough that it can promote the formation of multicellular units even if collective living puts cells in a disadvantage compared to solitary existence, see Fig 5.

For the described effect on the evolution of life cycles, the fragmentation costs must be paid not only in the course of multicellular development, but also be present in a similar form in the unicellular life cycle. However, paying fragmentation costs at each cell division appears to have no adaptive significance. Therefore, natural selection, instead of triggering the evolution of multicellular life cycles, may just lead to the removal of fragmentation costs themselves.

However, the life cycles of a number of unicellular species exhibit features which can be associated with some fragmentation costs. Our main example comes from the green algae genus *Nannochloris* containing only 7 species, all of them are unicellular [43]. Four of them are enveloped in a cell wall and reproduce by binary autosporulation: when the maternal cell divides, both daughter cells initially remain enclosed inside the cell wall, then they form cell walls of their own and break out, leaving the empty maternal cell wall behind. Each fragmentation in these unicellular species is accompanied by the loss of biomass (maternal cell wall) and therefore, is costly by our definition. Two more species reproduce by simple binary cell division and do not discard cell walls, their fragmentation is thus costless in our sense. The single remaining species reproduces by autosporulation, but undergoes two rounds of cell division before releasing four daughter cells, also paying a fragmentation cost in the form of a discarded cell wall. A phylogenetic analysis of this genus indicates that the ancestral form of life cycle is binary autosporulation [43, 44]. Hence, in the history of *Nannochloris* genus, most species preserved the original life cycle with costly binary fragmentation, some abandoned fragmentation costs, and one species developed multiple fission mode of reproduction.

Another example are *Volvocales* algae—a monophyletic group used to study the evolution of multicellularity, as it contains organisms with sizes ranges from one to tens thousands cells [45]. The unicellular member of the group is *Chlamydomonas reinhardtii*, where the cell is enclosed within a cell wall. Upon reproduction, the cell undergoes two or three rounds of division resulting in four or eight cells enclosed within the maternal cell wall. Later, these cells escape, leaving the empty cell wall [46, 47]. Other members of *Volvocales* are multicellular and pay noticeable costs of the group fragmentation. For example, in *Pandorina morum* (16 cells) and *Astrephomene gubernaculifera* (32 or 64 cells), the colonies are embedded in the common cell wall. Whenever a cell gives rise to a colony, the offspring colony breaks through the cell wall of the parent and escapes [48–50]. A logical conclusion of this process is that the maternal cell wall is completely abandoned, as reported for *A. gubernaculifera* [48]. Hence, in these multicellular species, the cell wall is not inherited either and constitutes an explicit fragmentation cost. An alternative to the development of a multicellular life cycle, would be not to produce a cell wall in the first place. Such species are not known among *Volvocales*. However, a number of cell wall less mutants have been reported for *C. reinhardtii* [47]. In laboratory conditions, these mutants have shown similar growth rates to the wild type [47]. Therefore, the cell wall is not essential for the survival of *C. reinhardtii* cells. Indeed, outside of the lab, the cell wall likely

has some adaptive significance for example by providing protection to the cell. Still, the evolutionary option to discard the cell wall and associated fragmentation costs is available to *C. reinhardtii*, but such a path was not taken.

Similar life cycles and fragmentation costs were observed in a range of other species, such as the bacterium *M. polyspora* [27] (see also Fig 1A), algae *Chlorella* [44], shizonts of Ichthyosporea [51], and among cyanobacteria of the *Pleurocapsales* order [30]. Therefore, the fragmentation costs considered in our model seem to be relevant to a wide spectrum of species.

In addition to the naturally existing fragmentation costs, these can emerge in the result of a mutation. Experimental evolution studies demonstrate several examples of these. In the settling experiment in yeast *S. cerevisiae*, the mutant cells are unable to separate after division, which leads to the formation of multicellular snowflake-structured clusters [6, 23, 24]. In the course of rapid evolution, these clusters evolved a primitive form of apoptosis leading to release of colony branches as offspring [6]. Such a reproduction mode is an example fragmentation with loss considered in our study.

In another life cycle experiment with bacteria *Pseudomonas fluorescens*, mutant cells overproduced cellulose at the cell surface, which made them stick together [7]. The fragmentation of the arising clusters likely requires significant efforts and it is safe to assume that it cannot happen for free. In addition, cellulose production itself is costly. Thus, in a well mixed environment, such a mutant loses the growth competition to the wild type [40]. Yet, in a non-mixed environment, cellulose-producing mutants form a mat on the air-liquid interface, gain exclusive access to both oxygen and nutrients, and ultimately outgrow the wild type [40]. Such a selective advantage arises only when the number of cells in the mat is sufficiently high. To get to that stage, the cells capable of producing the mat must pass through the initial phase of growth, where they have no advantage. Moreover, the return to the wild type is possible [7, 52], so abandoning the fragmentation costs altogether is an evolutionary option. Our model demonstrates that an increase in the group size under costly fragmentation is indeed an evolutionary plausible strategy, even if the size does not provide any benefits until a large amount of

**Table 1. List of species considered in this work and their life cycles.**

| Group | Species | Fragmentation mode | Fragmentation costs |
|---|---|---|---|
| *Nannochloris* | *N. bacillaris* [43] | $1 + 1$ | none |
| | *N. coccoides* [43] | $1 + 1$ | none |
| | *N. maculata* [43] | $1 + 1$ | cell wall loss |
| | *N. atomus CCAP 251/7* [43] | $1 + 1$ | cell wall loss |
| | *N. atomus SAG 14.87* [43] | $1 + 1$ | cell wall loss |
| | *N. sp. SAG 251-2* [43] | $1 + 1$ | cell wall loss |
| | *N. eucaryotum* [43] | $1 + 1, 1 + 1 + 1, 1 + 1 + 1 + 1$ | cell wall loss |
| *Volvocales* | *C. reinhardtii* [46, 47] | $1 + 1$ | cell wall loss |
| | *P. morum* [48–50] | $16 \times 1$ | cell wall loss |
| | *A. gubernaculifera* [48] | $32 \times 1, 64 \times 1$ | cell wall loss |
| *Bacteria* | *M. polyspora* [27] | $\sim 7 \times 1$ | maternal cell loss |
| | segmented filamentous bacteria [28] | $X + 1 + 1$ | cell wall and maternal cell loss |
| *Ichthyosporea* | *Ichthyophonus sp.* [51] | $> 20 \times 1$ | extracellular membrane loss |
| *Chlorella* | *C. vulgaris* [44] | $1 + 1, 4 \times 1, 32 \times 1$ | cell wall loss |
| | *C. kessleri* [44] | $1 + 1, 4 \times 1, 8 \times 1$ | cell wall loss |
| *Cyanobacteria* | *Pleurocapsales* [30] | $1 + 1, 4 \times 1, \ldots, > 20 \times 1$ | cell wall loss |
| *Fungi* | *S. cerevisiae* [23, 24] (experimental system) | $X + Y$ | cell loss |

cells is accumulated in a colony. The species discussed in our work, their life cycles, and sources of possible fragmentation costs are summarized in Table 1.

Concluding the discussion on unicellular species facing fragmentation costs, we would like to highlight that such organisms have two options to mitigate the costs: either to find a way to reduce the cost itself, or to alter their life cycle to a multicellular one to mitigate the impact of costs. Which of those two options is more plausible depends on the specific situation, and we see both paths taken in natural and experimental populations. Those species that adopt a multicellular stage in their life cycle gain the possibility to develop a wide range of beneficial adaptations available only to multicellular species, such as the division of labour.

## Supporting information

**S1 Text. Appendix.** Life cycles competition.
(PDF)

**S2 Text. Appendix.** Linear model of life cycles evolution.
(PDF)

**S3 Text. Appendix.** Forbidden fragmentation modes.
(PDF)

**S4 Text. Appendix.** Random environments.
(PDF)

**S5 Text. Appendix.** Binary fragmentation and (nearly) equal split are overrepresented in beneficial and unimodal environments.
(PDF)

**S6 Text. Appendix.** Only deterministic fragmentation modes can be evolutionarily optimal under any environment.
(PDF)

## Acknowledgments

We are grateful to David Rogers and Philippe Remigi for fruitful discussions and biological insights, to Jorge Peña for valuable detailed comments on the initial manuscript.

## Author Contributions

**Conceptualization:** Yuriy Pichugin.

**Formal analysis:** Yuriy Pichugin, Arne Traulsen.

**Investigation:** Yuriy Pichugin.

**Project administration:** Arne Traulsen.

**Software:** Yuriy Pichugin.

**Supervision:** Arne Traulsen.

**Visualization:** Yuriy Pichugin, Arne Traulsen.

**Writing – original draft:** Yuriy Pichugin, Arne Traulsen.

**Writing – review & editing:** Yuriy Pichugin, Arne Traulsen.

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
