## [Decision Letter · Decision Letter 0]

9 Apr 2020

Dear Dr. Pichugin,

Thank you very much for submitting your manuscript "Evolution of multicellular life cycles under costly fragmentation" for consideration at PLOS Computational Biology.

As with all papers reviewed by the journal, your manuscript was reviewed by members of the editorial board and by several independent reviewers. In light of the reviews (below this email), we would like to invite the resubmission of a significantly-revised version that takes into account the reviewers' comments.

The reviewers are split on this. One recommends minor revision, while the other recommends rejection. The latter raises a number of important concerns ranging from the novelty of the work in the context

of previous publications, through to the biological relevance of the work given the lack of appropriate data. I would be willing to consider a substantially revised version of the paper that fully takes into

account these comments (and a rebuttal letter setting out the changes).

We cannot make any decision about publication until we have seen the revised manuscript and your response to the reviewers' comments. Your revised manuscript is also likely to be sent to reviewers for further evaluation.

Sincerely,

Philip K Maini

Associate Editor

PLOS Computational Biology

Natalia Komarova

Deputy Editor

PLOS Computational Biology

The reviewers are split on this. One recommends minor revision, while the other recommends rejection. The latter raises a number of important concerns ranging from the novel of the work in the context

of previous publications, through to the biological relevance of the work given the lack of appropriate data. I would be willing to consider a substantially revised version of the paper that fully takes into

account these comments (and a rebuttal letter setting out the changes).

Reviewer's Responses to Questions

**Comments to the Authors:**

Reviewer #1: The authors point out that a great variety of life cycles exist in nature, although the factors affecting the evolution of life cycles and fragmentation modes are largely unknown. The authors previously found that costless fragmentation promotes the evolution of binary fragmentation modes, but fragmentation into more than two parts is common in biology. Costly fragmentation, however, is widespread in nature, and this article is on how costly fragmentation influences the evolution of life cycles.

In the model that the authors investigate, cells of an organism divide and stay together until they reach some maximum size, at which point the fragmentation mode determines how the organism divides into smaller groups of cells. The authors consider three types of fragmentation costs: delay, death risk, and cell loss. They find that costly fragmentation can favor many more types of fragmentation modes than costless fragmentation. The three types of fragmentation cost have the same set of forbidden life cycles. Binary fragmentation modes are promoted for the case of costly fragmentation. If fragmentation costs occur for complexes of different sizes, then benefits of forming complexes of multiple cells are not essential for evolution of multicellularity. The authors argue that the three types of fragmentation costs that they consider are applicable to a broad range of species.

This article is well-researched, and the topic of costly fragmentation has relevance for understanding the life cycles of many organisms. I consider this paper a great conceptual achievement. I recomend publication subject to minor revision.

I have some minor comments:

It might be helpful to cite the following articles:

Ghang, W. & Nowak, M.A. (2014). Stochastic evolution of staying together. J. Theor. Biol. 360: 129-136.

Olejarz, J. W. & Nowak, M.A.(2014). Evolution of staying together in the context of diffusible public goods. J. Theor. Biol. 360:1-12.

Kaveh K, Veller C, Nowak MA. Games of multicellularity. Journal of Theoretical Biology. 2016;403:143–158

Olejarz, J., Kaveh, K., Veller, C. & Nowak, M.A. (2018). Selection for synchronized cell division in simple multicellular organisms. J. Theor. Biol. 457.

Note also that the model for evolution of eusociality in Nowak, M. A., Tarnita, C. E. & Wilson, E. O. (2010). The evolution of eusociality. Nature 466(7310):1057-1062

is a mathematical analysis of life cycles with costly fragmentation. It might be useful to point out that very interesting connection.

page 5, middle: typo: "by dividing cell staying together" should be "by dividing cells staying together"

page 5, bottom: this might read better: "Units of size $i$ have death rate $d_i$, and cells in such a unit have division rate $b_i$."

page 6, middle: a comma is missing: "A partition is a way of decomposing an integer m into a sum of integers without regard to order, and summands are called parts [Andrews, 1998]."

page 7, top: "Next, we assume that the fragmentation of a unit comes at a cost for this unit. In other words, if the unit continues to increase in size instead of fragmenting, its expected net biomass in the short run will be larger." Could that statement be clarified?

page 7, middle: typo: "maximises" should be "maximizes"

page 7, bottom: typo: "where T it the fragmentation delay" should be "where T is the fragmentation delay"

page 8, caption of Figure 2: typo: "and remaining four are split" should be "and the remaining four are split"

page 11, top: typo: "see Appendix A.2 for details of the matrix construction." should be "(See Appendix A.2 for details of the matrix construction.)"

page 13, caption of Figure 3: typo: "and is thus called forbidden" should be "and are thus called forbidden"

page 16, caption of Figure 4: typo: "Fragmentation by binary and (nearly) equal split partitions are likely to evolve in random environments." should be "Fragmentations by binary and (nearly) equal split partitions are likely to evolve in random environments."

page 18, caption of Figure 5: typo: "found her have" should be "found have"

page 19, bottom: typo: "these modes delivering" should be "those modes delivering"

page 21, middle: typo: "both daughter cell" should be "both daughter cells"

page 21, middle: typo: "leaving empty maternal cell wall behind" should be "leaving the empty maternal cell wall behind"

page 22, top: typo: "cell give rises to the a colony" should be "cell gives rise to a colony"

page 22, bottom: typo: "which lead to the formation" should be "which leads to the formation"

page 23, top: typo: "which make them stick together" should be "which made them stick together"

page 23, middle: "Which of two" should be "Which of those two options"

page 23: The last paragraph should be either deleted or rewritten.

Reviewer #2: The paper is a theoretical investigation of how simple multicellular organisms evolve. The number of cells in these multicellular units (stochastically) grows and upon reaching a certain size, these units fragment into smaller units. The authors are interested in the most efficient way to do this. Or rather, the effects of the cost of fragmentation on the most efficient way. In order to do this they study rate equations describing the mean behavior.

My problem with the paper is that it falls somewhere between useful and beautiful. It is very far from experimental testing, nothing is known, so many parameters are assumed. To get answers one needs to know these parameters. As they write: "Unfortunately, such experimental data is not available, neither for natural nor for experimental populations." Hence the authors assume random parameters from some arbitrary distributions. And the results of course cannot be compared to data either.

And this would be OK if the model was pretty, which is subjective of course, but I find it cumbersome. The rules are quite arbitrary; one could similarly set up models with very different rules, like fragmenting at some rate at each size, fragmenting randomly, and so on.

The authors already published a conceptual paper Pichugin et al 2017, building upon Tarnita et al 2013. That paper made the point that there could be optimal fragmentation modes, and suggested that "in the field and in the laboratory" this could be studied. Last year a stochastic version of the model was published: Pichugin et al 2019.

In summary, I am not convinced that different versions and aspects of this model is worth being published in this Journal without meaningful links to biology.

**Have all data underlying the figures and results presented in the manuscript been provided?**

Reviewer #1: Yes

Reviewer #2: Yes

PLOS authors have the option to publish the peer review history of their article (what does this mean?). If published, this will include your full peer review and any attached files.

Reviewer #1: Yes: Martin Nowak

Reviewer #2: No
---

## [Decision Letter · Decision Letter 1]

6 Aug 2020

Dear Dr. Pichugin,

Thank you very much for submitting your manuscript "Evolution of multicellular life cycles under costly fragmentation" for consideration at PLOS Computational Biology. As with all papers reviewed by the journal, your manuscript was reviewed by members of the editorial board and by several independent reviewers. As Reviewer 1 had only minor concerns but Reviewer 2 recommended Reject we sent it back to Reviewer 2. Unfortunately, despite several reminders, Reviewer 2 did not respond, so we asked a third Reviewer to assess the

whole situation given the previous reviewers. This reviewer is very positive but raises some issues. We would be very happy to consider a revised version that takes this comments into account.

Sincerely,

Philip K Maini

Associate Editor

PLOS Computational Biology

Natalia Komarova

Deputy Editor

PLOS Computational Biology

[LINK]

Reviewer's Responses to Questions

**Comments to the Authors:**

Reviewer #3: Review: Evolution of multicellular life cycles under costly fragmentation

The authors present an an analysis of how costs associated with the fragmentation of multicellular complexes may drive the evolution of particular fragmentation modes – for instance whether to cleave in to two equally sized parts, produce unicellular propagules or spilt into a number of unicells. The identification of conditions that lead these latter two results are particularly interesting, arising as they do thorough extensions of the authors’ previous 2017 work, where fragmentation costs were only briefly (and incompletely) touched upon. In the conclusion in particular, I believe the authors put together a solid case that different fragmentation costs could lie behind the variety of fragmentation modes observed in nature. From a theoretical standpoint, I also found the observation that these costs could actually drive the evolution of multicellularity itself very intriguing. On first reading I felt like this was likely be a purely theoretical curiosity. However following the authors’ discussion of reproduction in the Volvocales algae, I’m actually thoughtful of paying this more heed.

I think this is a thoughtful and thorough article, well-worthy of publication in PLOS Computational Biology. As always however, I believe there are some changes that the authors could make to improve the manuscript still further.

The initial three suggestions I believe in some sense may address the concerns of Reviewer 2.

First, for the random environments the authors choose b_i and d_i sampled independently from the uniform distribution U(0,1). As Reviewer 2 states “Hence the authors assume random parameters from some arbitrary distributions. And the results of course cannot be compared to data either.” While I do not share Reviewer 2’s concern about data in a quantitative sense, it seems more biologically relevant to consider three subcases for b_i; b_i strictly increasing, b_i strictly decreasing, and a unimodal distribution for b_i (and simultaneously the converse scenario for d_i) . Importantly, it seems biologically unlikely that b and d are multimodal, as is possible in the current implementation.

The authors have already considered the second of these scenarios in Section 4.5, so there’s no extra work there. The case of strictly increasing b_i should be similarly straightforward. My guess would be that in this condition the “Other” fragmentation modes would be unlikely. The case of unimodal b_i would still allow for a formidable parameter space. A manageable subcase to look at might be b peaked in the middle at b_10. My guess here would be that we might see an increase in the maturity size selected for compared to the case with no costs, similar to the current results in Section 4.5.

Second, in response to Reviewer 2’s concerns about data, the authors state “Predictions from our model are ready to be tested once experimental data are available, but we are not experimentalists and not in a position to conduct such experiments (as probably most mathematical biologists). In the revised manuscript, we reflect this aspect of our work in the discussion”. I believe the discussion tackles this very well, but suggest that the predictions of the model could be even more succinctly stated for use by experimentalists by using a table. This could be included in addition to the discussion (which could be kept as is) and could essentially list each species described by the authors, its empirically observed fragmentation mode, and the model predictions for scenarios under which this mode might be selected for. The table would then serve as a handy reference for experimentalists interested in testing these hypotheses.

Thirdly, Reviewer 2 is concerned that the current paper is perhaps incremental “The authors already published a conceptual paper Pichugin et al 2017 … I am not convinced that different versions and aspects of this model is worth being published in this Journal ...”. Given this it appears that it may not be emphasised clearly enough which predictions of the current model are novel with respect to the 2017 paper. I would suggest periodic reminders in the results section. To avoid repetition I would move the line 267-268 “Only binary fragmentations can be evolutionarily optimal under costless fragmentation, T = R = L = 0 [Pichugin et al., 2017]” (it’s not necessary to address the evolutionary dynamics in that section) and place it in a paragraph before L316 noting that nearly equal split, seeding, and other are not possible in the costless case. The following paragraphs could then emphasise how each fragmentation cost extends the life-cycles of the costless case – e.g. following line 321 “Thus delay fragmentation costs extend the possible life cycles to equal splits relative to the costless case”.

My other comments are more minor/unrelated to previous reviews.

5) I was surprised to see discussion of the unicellular bottleneck omitted. I am not suggesting that this should be dealt with in a modelling sense, but it certainly should be discussed. It is likely that this is a strong factor driving a transition from equal-part fragmentation modes to those involving the production of unicells.

6) Section 3.2 could be improved by explicitly giving an example with a given \\pi(\\kappa) and m, and emphasising these as evolutionary parameters. This might also help make the life-cycle definitions (e.g. 1+1 lifecycle) more clear.

7) L149 - “The fragmentation cost is another external factor of our model” - would it be more natural to think of this as a physiological restriction, based on the evolutionary history of the organism under consideration?

8) Fig 2 could reference Eqs.(1-2) for the b/d costs.

9) Figure 3 – in the “nearly equal split”, 1+1+1+…1 is included, yet this seems by far the more interesting example in this category biologically, and it also appears in particular under various scenarios. Would it be worth considering this as a separate case?

10) L133 fast → quickly

**Have all data underlying the figures and results presented in the manuscript been provided?**

Reviewer #3: Yes

PLOS authors have the option to publish the peer review history of their article (what does this mean?). If published, this will include your full peer review and any attached files.

Reviewer #3: **Yes: **G.W.A. Constable
---

## [Decision Letter · Decision Letter 2]

28 Sep 2020

Dear Dr. Pichugin,

We are pleased to inform you that your manuscript 'Evolution of multicellular life cycles under costly fragmentation' has been provisionally accepted for publication in PLOS Computational Biology.

Best regards,

Philip K Maini

Associate Editor

PLOS Computational Biology

Natalia Komarova

Deputy Editor

PLOS Computational Biology

Reviewer's Responses to Questions

**Comments to the Authors:**

Reviewer #3: Great to see the suggestions implemented in this final version - I have no further recommendations for the scientific content of the paper.

Following the spot of the small bug in the production of figure 5, it may be wise to upload the simulation and figure code to a repository, and to provide a link in the paper along with the summary data for the figures.

Finally, just one small typo - L337 (birth rate increase with size, while death rate decrease) -> (birth rate increases with size, while death rate decreases)

**Have all data underlying the figures and results presented in the manuscript been provided?**

Reviewer #3: **No: **I can't see a link to the data for Figure 3B, 4 or 5 - however I may have overlooked this.

PLOS authors have the option to publish the peer review history of their article (what does this mean?). If published, this will include your full peer review and any attached files.

Reviewer #3: **Yes: **George WA Constable

---

## [Editor Report · Acceptance letter]

1 Nov 2020

PCOMPBIOL-D-20-00157R2 

Evolution of multicellular life cycles under costly fragmentation

Dear Dr Pichugin,

I am pleased to inform you that your manuscript has been formally accepted for publication in PLOS Computational Biology. Your manuscript is now with our production department and you will be notified of the publication date in due course.

With kind regards,

Nicola Davies
